# A cluster randomised controlled trial investigating the effectiveness of the 'Support group' intervention in primary schools in Norway: A study protocol

Lisbeth Valla[1,2]*, Bente Sparboe-Nilsen[1,3], Lisbeth G. Kvarme[1], Therese Haugerud Bjerketveit[1], Milada Hagen[1], Brita Askeland Winje[1]

1 Department of Nursing and Health Promotion, Faculty of Health Sciences, Oslo Metropolitan University, Oslo, Norway, 2 Section for Infant Mental Health, Regional Centre for Child and Adolescent Mental Health, Eastern and Southern Norway, Oslo, Norway, 3 Faculty of Medicine and Health, Orebro University, Sweden

* lisval@oslomet.no

## Abstract

Bullying is a pervasive public health issue that significantly affects the mental and physical well-being of children, often leading to long-term consequences that persist into adulthood, such as mental health disorders, social isolation, and economic challenges. Despite numerous interventions implemented in schools, bullying remains a persistent problem. The Support Group intervention, based on a solution-focused approach (SFA), aims to address bullying by enhancing peer support and empowering children within the school environment. This study seeks to evaluate the effectiveness of this intervention in reducing bullying and improving class environment, mental health, quality of life, and self-efficacy among 5th-7th grade Norwegian schoolchildren compared to usual care. This cluster randomized controlled trial (RCT) will be conducted in 26 primary schools across three municipalities in central-eastern Norway. Schools will be randomized to either the intervention group, which will implement the Support Group intervention, or the control group, which will continue with usual care based on national guidelines. Data will be collected at baseline, post-intervention, and at 6- and 9-month thereafter. A parallel process evaluation will be conducted to assess the fidelity and quality of the intervention's implementation and to identify factors associated with its success. The intervention will be compared with usual care using generalised mixed models for repeated measures. This study will provide crucial insights into the effectiveness of the Support Group intervention in reducing bullying and promoting mental health among schoolchildren. The findings are expected to inform best practices in bullying prevention and contribute to the development of more effective, evidence-based interventions in school settings. The trial outcomes will be pivotal in shaping future strategies for creating safer and more supportive school environments in Norway. The study is registered in Clinical trial registration: Trial registration (NCT) 06578260.

**Data availability statement:** No datasets were generated or analysed during the current study. All relevant data from this study will be made available upon study completion.

**Funding:** The author(s) received no specific funding for this work.

**Competing interests:** The authors have declared that no competing interests exist.

## Background

Bullying is defined as repeated aggressive behaviour characterised by an observed or perceived power imbalance between the victim and perpetrator [1]. A newer definition includes social exclusion and points out exclusion as a possibility in all social groups and contexts. The risk of being judged unworthy to belong to a social group includes feelings of being misunderstood, not seen, socially threatened and deprived of dignity[2]. Bullying is considered to be a significant public health problem with both short- and long-term physical and social-emotional consequences for the victim [3]. The negative consequences of bullying are pervasive, and bullying exposure in childhood is also associated with poor mental and physical health, lack of social relationships, economic hardship, and decreased quality of life in early adulthood and midlife [4–6]. A good psychosocial environment in school is paramount in reducing the burden associated with mental health disorders in young people [7]. Hence, the government of Norway have maintained that in accordance with regulation for health promotion and prevention in the school health services, the school nurse and school staff should collaborate to create a good and healthy environment for the children. If there is any indication of bullying, school nurses, in collaboration with school staff, are obliged to intervene early at class-, group- and/or individual level [8]. Although appropriate actions and practice to prevent bullying and create a good environment within the schools exist [9,10], bullying is still reported to affect around 14.7% of children and young people in Nordic countries [11], and six percent of children in Norway [12]. The Norwegian student survey for 2023 shows a clear increase in elementary school students experiencing bullying. In the 7th grade, 13% responded to being subjected to bullying, while in the 10th grade, the percentage was 11 percent [NOU 2023:1 Regjeringen.no].

The growing awareness of bullying has led to the implementation of different school-based anti-bullying programs in the last 20 years [13]. Some meta-analyses [3,14–18] have reported small to moderate effectiveness of anti-bullying programs, with a mean decrease of approximately 20% in bullying rates. However, variation in effectiveness exist due to methodological designs, program types, and geographic regions, which indicate the need for additional research [3,17–19]. Gaffney et al, [2021] evaluated the effectiveness of over 80 school-bullying intervention programs and indicated several intervention components that were crucial to intervention outcomes [18]. For example, informal peer involvement like group discussions or role-playing activities reduced bullying perpetration by about 12.5% and bullying victimisation by 9%.

The Support group intervention is a peer group program using a solution-focused approach [SFA], which is based on children's own goals, strengths and resources. The SFA method has previously shown to be an effective treatment strategy for a range of behavioral and psychological outcome [20,21], and is implemented in many schools and school health services. This project will investigate the effectiveness of a Support group intervention utilizing the SFA method for reduction of bullying. The intervention is designed especially for schoolchildren in primary school. In the

Support group intervention, school nurses, teachers and students collaborate in a systemic way within the school setting to improve a bullied child's situation through enhancing levels and quality of peer support. Limited research has explored the effect of anti-bullying intervention based on the SFA method. Kvarme et al [2016] found in quality interviews among 19 schoolchildren aged 12–13 years that support groups contributed to the cessation of bullying. The improvements remained 3 months later [22].

## Pilot study, 2020–2022

The present study is based on a pilot study which tested the feasibility and acceptability of the support group intervention in seven primary schools in South- East Norway [23,24]. The pilot study ended in December 2022. The non-profit organization "*Adults for Children*" [Voksne for Barn] collaborated with researchers in the implementation of the intervention. A total of 20 social teachers and/or public health nurses and 235 children aged 9–12 years participated in the study. The pilot study had a quasi-experimental design, and both qualitative and quantitative data were collected. Two articles reporting on qualitative aspects have been published [23,24]. Overall, both teachers, public health nurses, parents and school children were satisfied with the intervention. Some children reported that they felt stronger, safer, and happier because of the support group, while other children said that the intervention improved the atmosphere in the classroom. The support group was highlighted as a practical work tool by school nurses [24].

## Theoretical approach

In recent years, bullying has to a greater extent been seen as social processes that must be addressed with systemic approaches instead of individual measures [25]. Bullying is not merely seen as a dyadic problem between peers and a lonely or excluded child; instead, it is acknowledged as a group phenomenon, arising within a social context where various factors play a role in promoting, sustaining, or alleviating such behaviour [26]. The social-ecological system theory, based on Bronfenbrenner's ecological model, contends that the lonely child is part of complex, interrelated system levels that place them at the centre and moving outwards from the centre to the various systems that shape the individual. These levels encompass micro- [peers, family, community, and schools], meso- [interactions between components of the microsystem], exo- [social context], macro- [social, cultural, and political contexts] [27,28]. At the microsystem level, the Support Group intervention enhances peer relationships and creates a supportive social network to address the child's immediate social context. The mesosystem is engaged by strengthening collaboration between school staff and families, promoting a shared commitment to bullying prevention. At the exosystem level, the intervention aligns with school policies to integrate systemic approaches to bullying prevention. The macrosystem is indirectly influenced by raising awareness of bullying as a broader societal issue, fostering inclusivity and social responsibility. Mediators such as improved peer support, enhanced empowerment through self-efficacy, and the creation of a positive classroom environment are critical pathways through which the intervention achieves its effects. These mechanisms are hypothesized to lead to proximal outcomes, such as reduced bullying incidents and improved peer relationships. Over time, these proximal changes are expected to translate into distal outcomes.

In the area of bullying, this perspective is used to understand how characteristics of the individual child interact with environmental contexts or systems.

## Study objectives

The primary study objective of this project is to investigate the effectiveness of a support group intervention using a solution-focused approach in reducing bullying, enhancing mental health, improving quality of life, and increasing general self-efficacy among school children in 5$^{th}$-7$^{th}$ grade.

We hypothesize that the awareness of bullying created by the intervention, and the empowerment of children as contributors to the solution, will generate positive ripple effects benefiting all children in intervention schools. Furthermore,

 

we hypothesize that children with a peer support group will exhibit lower levels of bullying, improved mental health, better quality of life, and increased general self-efficacy after the intervention compared to their baseline levels. We expect these effects to persist even after 6 and 9 months.

Aligned with the cluster RCT we will undertake a parallel process evaluation to assess the extent of the intervention coverage, whether the intervention was implemented according to the protocol and to identify factors that hinder or aid the implementation of a support group intervention. For the process evaluation we will use the framework presented by the Medical Research Council guidelines focusing on the implementation [what is implemented and how], the mechanisms of impact [how does the delivered intervention produce change], and the context [how does the context affect implementation and outcomes][29]. The outline of the process evaluation is described later in the protocol.

## Research questions

### Primary research questions

**pRQ1:** To what extent do children in intervention schools report a reduction in **bullying** following a peer support group intervention compared to children in control schools receiving usual care, at intervention end and after 6- and 9 months?

**pRQ2:** To what extent do children in intervention schools report higher **quality of life** following a peer support group intervention compared to children in control schools receiving usual care, at intervention end and after 6- and 9 months?

**pRQ3**: To what extent do children in intervention schools report improved **classroom environment** following a peer support group intervention compared to children in control schools receiving usual care, at intervention end and after 6- and 9 months?

**pRQ4**: To what extent do children with a peer support group report a reduction in **bullying** at the end of the intervention compared to their pre-intervention baseline level?

### Secondary research questions

**sRQ5**: To what extent do children with a peer support group report improved **mental health** at intervention end compared to their pre-intervention baseline level?

**sRQ6**: To what extent do children with a peer support group report better **quality of life** at intervention end compared to their pre-intervention baseline level?

**sRQ7:** To what extent do children with a peer support group report increased **general self-efficacy** at intervention end compared to their pre-intervention baseline-level?

**sRQ8**: To what extent does the effects as measured in R1-R7 persist 6- and 9 months after intervention end compared with pre-intervention baseline level?

**sRQ9**: To explore the recruitment rate and factors associated with non-participation, on school, provider, and child level.

**sRQ10**: To describe whether the intervention was delivered according to protocol in terms of quality, quantity, adaptions and variations across schools and time.

**sRQ11**: To describe the intervention acceptability from children's, teachers, school nurses and residents' perspectives.

**sRQ12**: To investigate barriers and facilitators to intervention delivery and the mechanisms of impact.

## Methods

### Study design

This is a cluster randomized controlled trial of children in 5th to 7th grade in Norwegian primary schools with randomization at school level. All outcomes will be assessed at baseline, after the intervention is delivered and at 6 and 9 months thereafter. The cluster design is well-suited due to the significant risk of contamination between the groups with a regular

RCT design, e.g., it is not feasible to implement both the intervention and carry on with the usual care in the same school. A SPIRIT schedule and overview of the study design is depicted in "Fig 1". The status and timeline of the study is shown in "Fig 4". The recruitment period for this study started on 9th November 2024 and will end on 30th December 2026.

This protocol has been developed following the Standard Protocol Items: Recommendations for Interventional Trials [SPIRIT] checklist [2]

The study flow is presented in "Fig 2". Intervention schools will implement the SFA-model for reducing bullying, whereas the control schools will offer usual care according to national guidelines. The intervention is at two levels; [i] training of school nurses and school staff and raising awareness about the SFA-model among parents and students; for this part the outcomes will be measured at cluster-level, and [ii] implementing the SFA-model on individual participant level; for this part the outcomes will be measured on individual level in the intervention cluster only. The recruitment process, the intervention and the outcomes measured are described below. In this study, the standard school-based measures for bullying prevention, as recommended by national guidelines, are used as the comparator in the control group. This choice was made to allow for the comparison of the effectiveness of the Support Group intervention with the current standard practices in Norwegian schools. By using a control group that continues with usual care, we can measure the relative impact of the Support Group intervention and evaluate whether it yields better outcomes than the interventions

| | Enrolment | Allocation | Post-allocation (Intervention) | | | |
|---|---|---|---|---|---|---|
| **TIMEPOINT\*\*** | **-t₁** | **0** | **t₁** | **t₂** | **t₃** | **t₄** |
| **ENROLMENT:** | | | Baseline | Immediate post intervention | 6 months | 9 months |
| **Eligibility screen** | x | | | | | |
| **Informed consent** | x | | | | | |
| **Allocation** | | x | | | | |
| **INTERVENTIONS Supportgroup:** | | | x | x | x | x |
| **Control (Usual care)** | | | x | x | x | x |
| **ASSESSMENTS:** | | | | | | |
| Primary Outcome (Bullying) | | | ←——————————————→ | | | |
| Primary Outcome (Quality of life) | | | ←——————————————→ | | | |
| Primary Outcome (classroom environment) | | | ←——————————————→ | | | |
| Secondary Outcomes (Mental Health, Self-Efficacy) | | | ←——————————————→ | | | |

**Fig 1. SPIRIT diagram.** Schedule of enrolment, interventions, and assessments for the study. []||[]|[]|[].

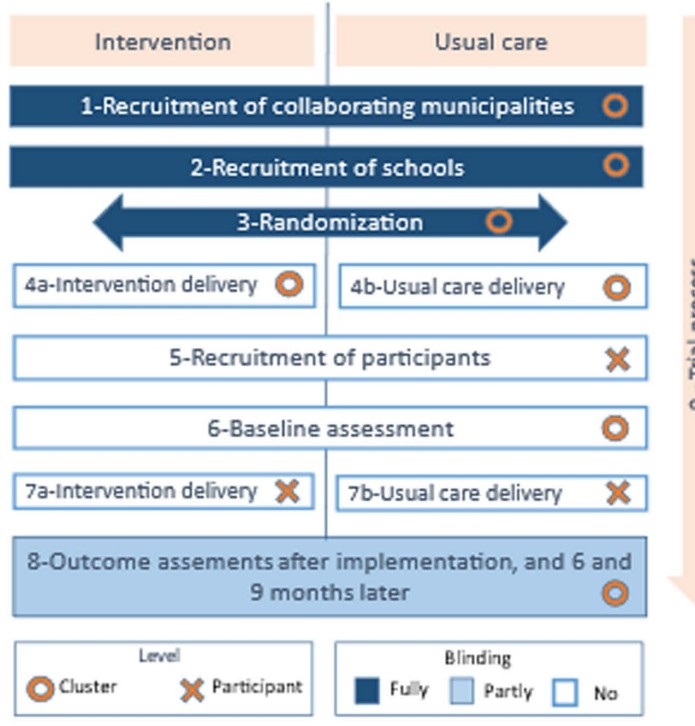

**Fig 2. Study flow for the Cluster randomized controlled trial.**

already in place. This comparison with usual care is crucial for determining the practical value of implementing the Support Group intervention as a broader measure in the school system. In this study, blinding will be limited due to the nature of the intervention. Participants and school staff involved in delivering the intervention, and conducting the baseline and follow-up assessments cannot be blinded due to the visible nature of the intervention (Support Group). However, outcome assessors and data analysts will remain blinded to group assignments to reduce bias in the evaluation of results. Any unblinding, if necessary, will follow strict guidelines to ensure it does not affect the trial's integrity.

The target population for the study are all school children in 5th-7th grade in primary schools in three municipalities (Asker, Nittedal and Nordre Follo) in the central-eastern part of Norway. Primary schools with more than 200 students and with a school nurse appointed in at least 50% position are eligible for the study. Eligibility for having a support group established is based on the subjective experience of loneliness by the individual child and is defined as a negative emotional state caused by the discrepancy between the desired and actual social relationships, in line with Peplau et al [30].

## Clusters and randomization procedure

Since decisions about bullying interventions are typically made at the municipal level, we will begin the recruitment process of schools with enrolment of selected municipalities, as illustrated in Fig 2. In selected municipalities consenting to participate, we will randomly select 26 primary schools from the pool of eligible primary schools in these municipalities and invite them to participate. The illustration of the recruitment process of the primary schools in the included municipalities is shown in "Fig 3".

We will continue enrollment until the pre-identified number of enrolled schools is reached. The enrolled schools will be sorted by size and then block-randomized using a 1:1 ratio to either the intervention arm (hereafter referred to as intervention schools) or the control arm (hereafter referred to as control schools). A statistician not involved in the study will carry

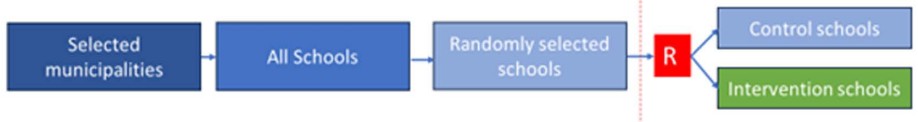

**Fig 3. Illustration of the recruitment process of primary schools in the included municipalities in Norway.**

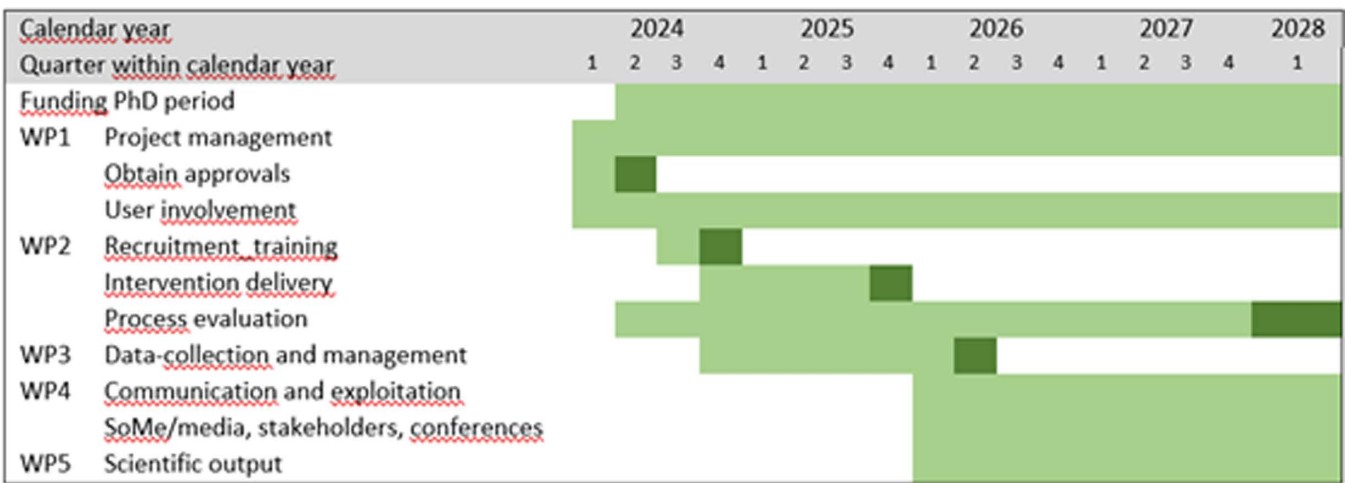

**Fig 4. Study progress plan with main activities and milestones.** *Darker color indicates that a task should be completed.*

out the randomization procedure. After randomization, the schools will be informed about their assigned study arm and will receive more detailed information accordingly

## Sample size calculation

*School level.* Based on the available literature, we anticipate that 10% of children in 5th -7th grade will report bullying [11,12]. This proportion is expected to be reduced to at least 6% in intervention schools' post-intervention. With a significance level of 5% and power of80%, we will need 721 children in each study arm. Accounting for additional variation due to clustering, we would need 793 in each arm. We anticipate that there will be three classes in each school with about 30 children in each class. Thus, we aim to include 26 schools (13 for each study arm) to ensure that the study is sufficiently powered. *Individual child level*: The quality-of-life inventory KIDscreen-27 [31] has 5 domains and is scored from 0–100. For the domain "social support and peers", we anticipate a change of at least 3 points from baseline to 9 months [32]. Further, assuming a correlation of 0.7 to account for statistical dependencies [as the same child is assessed several times], and keeping significance level to 5%, we would need 55 children experiencing loneliness. To account for possible dropouts, we add additional 10% participants, i.e., we need to enroll 60 children reporting feeling of loneliness.

## Study procedure

Essential information about the intervention, including the recruitment process and the continuation and follow-up phases will be shared with the municipality, students, parents, teachers and school nurses involved in the project. Written information about the option for support groups for children encountering challenges related to bullying at school will also be provided to parents. To enhance participant retention, the trial will implement regular follow-up meetings and reminders.

These will include scheduled consultations with school staff and ongoing check-ins to ensure engagement. For participants who discontinue or deviate from the intervention protocols, all relevant outcome data, including baseline and follow-up assessments, will continue to be collected to the extent possible. This approach ensures that valuable data are still obtained even if the intervention is not completed as intended.

### Training

*Intervention Schools*. A group of two to three teachers or school nurses will participate in a three-day training program centered around the solution-focused approach and the support group model. This training will be led by *Adults for Children*, an organization possessing extensive expertise in implementing interventions within school settings. Following the training, the peer support group intervention will be initiated as soon as possible. *Control Schools*: In the control schools, schools and preventive and health promotion services will be provided as usual and according to the guidelines set forth by the Norwegian Directorate of Health [8]. This may include structured anti-bullying plans, teacher-student relationship building, parental collaboration, teacher supervision, student counselling. Specialized teams may assist in severe cases

### Support group Intervention

A comprehensive implementation manual has been developed for this intervention, and key steps are presented in Table 1. The first initiative for a support group may come from the child itself or other individuals concerned about the child's well-being. Initiating a dialogue is the first step in building trust and determining if a support group could help. The child, together with the teacher or school nurse, will assess whether the support group is suitable for them. The school nurse and teachers will evaluate the child's situation carefully before recommending a support group, and if necessary, they will explore alternative follow-up options, particularly for children with complex challenges.

In **step 1**(the first meeting with the child) the teacher or school nurse have a solution-focused conversation with the child. In the conversation, the teacher or school nurse will reassure the child that they will do their utmost to help. They acknowledge and validate the child's experience of the situation and focus the conversation on possibilities, hope, strengths, and resources. The teacher or school nurse assist the child in articulating what is working and how they can contribute to achieving the desired situation. Scaling questions or the miracle question can be used to help the child articulate their thoughts more concretely. The goal in the first meeting is also to identify peers to be invited as support group members, and they ask the child who they enjoy spending time with, who they are friends with, who they wish to know better, and who they feel unsafe around or who has caused negative or indirect negative experiences. It is crucial to agree with the child on what will be communicated to their parents, teacher, and other staff, as well as what will be conveyed to the peers invited to the support group. During this first solution-focused conversation, the child will be encouraged to reflect on: How will they notice the situation improving, and how others will notice the improvement. The child will be asked to observe all small improvements, even the slightest positive changes to the next meeting. The child is individually followed with weekly consultations with the teacher or school nurse, using an SFA-counselling approach.

**Table 1. Key steps in Supportgroup.**

| The solution-focused counseling approach (SFA) is used in each step |
| --- |
| **Step 1:** *Meeting with the child who needs help.* Create hope and identify participants for the support group. |
| **Step 2:** Conversation with Parents and Establishing a Support |
| **Step 3:** Solution-Focused Conversation with Potential Support Members |
| **Step 4:** *Follow up meeting with the bullied/excluded child.* In line with SFA, look for progress, create motivation and faith that he/she can contribute to make a change. |
| **Step 5:** *Follow up meeting with the support group.* Give positive feedback and confirm all efforts made. The progress continues until the situation improves. |

**Step 2** involves a conversation with the child's parents and the establishment of the support group. In this step, participants for the support group are identified, and the child's parents are involved and provided with information about the intervention. While the child experiencing bullying will not join the group, they help select 5–7 peers and decide what details can be shared with the group. Participation in the support group is voluntary for peers. The teacher or school nurse determines the timing and location.

In Step 3, the teacher or school nurse meets with potential support group members and invites them to join promptly, ideally on the same day or the day after the initial conversation. This quick action reinforces to the child that the school is taking immediate steps.

Support group members can be invited individually during class or through a brief meeting. For example: "Hi (Teacher's Name), may I borrow (Name) and (Name)? I need their help with something. Would you join me to help?" During the meeting with the support group, the focus is on asking if they are willing to help a peer in need. The teacher or school nurse emphasizes the school's responsibility for everyone's safety and encourages their support. The conversation avoids assigning blame or discussing personal relationships, such as the causes of exclusion or bullying.

The discussions focus on:

• What can be done to support the child in question and improve their well-being

• What difference these actions can make for the child

• What changes the group members themselves may notice

The group brainstorms simple actions to support their peer in the coming week. Common suggestions include saying hello, sitting together, or joining in activities. The teacher or school nurse praises all suggestions and helps the group reflect on the positive impact their actions can have. These discussions typically last 10–20 minutes.

Studies indicate that the situation often improves rapidly, and in many cases, 3–5 weeks is sufficient [21–24]. Participation is voluntary, and the children decide what actions they wish to take until the next meeting. This transforms the group into a catalyst for improving a child's situation through small initiatives, using fellow students as a resource for change [24]. Peers are encouraged to concentrate on improvements and will receive acknowledgement for their progress. In **Step 4**: the teacher or school nurse have a solution-Focused Follow-Up Conversations with the bullied child. This step involves an individual follow-up with the child, where the focus is on small improvements, and were the the child reflecting on effects. The teacher/school nurse and the child reflecting together on the perceived differences so far, and the teacher or school nurse praising the child's efforts and contributions.

In **Step 5** the teacher or school nurse have a solution-Focused Follow-Up Conversations with Support members. There are no obligations regarding the suggestions made by members. Simply ask open-ended questions about their actions and praise all efforts. If no actions were taken, ask what they might consider doing next.

Participants may discontinue or modify their involvement in the Support Group intervention if they experience distress, significant improvement or worsening of their situation, or upon request by the participant or their guardian. During the trial, participants are permitted to receive any standard school health services and psychological support provided by the school. While we will not prohibit participants from receiving other interventions, we will monitor and document any additional bullying prevention or psychological interventions they may receive during the trial. This is to ensure that we have full oversight of all concurrent interventions, allowing us to assess their potential impact on the outcomes of the Support Group intervention.

## Monitoring

No formal data monitoring committee will be established for this trial, as the nature of the intervention (support group for schoolchildren) poses minimal risk to participants. Instead, the research team will monitor the trial's progress, including adherence to protocols and any reported adverse events. Regular reports will be reviewed by the principal investigators, and any issues requiring escalation will be addressed in accordance with ethical guidelines.

## Strategies to improve adherence to intervention schools

To ensure adherence to the Support Group intervention, regular follow-up meetings will be conducted between researchers and school staff. Monitoring will include weekly check-ins to track participant engagement and address any issues that may arise.

## Instruments and measures

Self-reported experience of being bullied will be assessed with Olweus questionnaire [26]. It comprises of 14 items, each scored from 0: not being bullied at all, 1: very rarely, 2: 2–3 times a week and 4: once or more times a week. The total score is constructed as a sum of the 14 items, thus ranging from 0–42. The questionnaire has been validated in a Norwegian setting [32].

Quality of life (*HRQoL)* will be measured using the Norwegian version of the Kidscreen 10 self-report questionnaire [33].. The Kidscreen 10 is a validated in a Norwegian population [31] considered reliable, short and multidimensional measure of generic HRQoL in children and adolescents. It comprises 10 items scored on a Likert's scale from 1 to 5. The total sumscore ranges from 10–50 with higher levels indicating a better heath related quality of life. The SDQ will be used to assess possible self-reported mental health problems. The SDQ consists of 20 items divided into four dimensions: emotional problems, peer problems, conduct problems and hyperactivity-inattention, which can be sum up into a total difficulties score. Higher scores indicate more severe problems. Each SDQ-item is scored on a Likert's scale ranging from 0: "not true", 1: "somewhat true", 2: "certainly true", negatively worded items are reversed. The SDQ has been translated to Norwegian and validated in the general population of Norwegian children [34].

Level of self-efficacy will be assessed using the General self-efficacy scale. It comprises 10 items, each scored on a Likert's scale from 1: completely wrong, 2: quite wrong, 3: partially correct, 4: completely correct. The total score is constructed as a sum of all the items thus ranging from 10–40 with higher scores indicating higher levels of self-efficacy. The score has been validated in a Norwegian setting [35]. Additionally, The pupil-survey (Elevundersøkelsen, UDIR) comprises 30 questions covering areas as enjoyment, being bullied, friends and class environment. The questions are scored differently depending on the questions. The questionnaire is divided into 5 domains and in general higher total domain scores indicate a positive assessment. The questionnaire has not been validated, however it has been wildly used in Norwegian schools. It also covers basic socio-demographic variables such as age and gender of the responders.

Table 2 shows overview of planned studies, outcome measures and time of measurement.

## Process evaluation

The study will assess the implementation processes using both quantitative and qualitative data. We thereby are following the recommendation when it comes to developing and evaluating complex interventions [29,37] where process evaluation plays an important role. Process evaluations may help to explain divergences between expected and observed outcomes as well as helping to explain how context influences outcomes. Moreover, process evaluation yields insights that might assist the implementation process [29]. Quantitative methods will assess intervention fidelity [Adherence, dosage, quality of delivery], and recruitment of schools and school children. To assess fidelity of indicated actions, each school [intervention] will be asked to complete a termly checklist which captures the extent of groups and to which bullying incidents have been dealt with in line with the Support group strategies, procedures, and documentation. All schools [intervention and control] will be asked at recruitment to complete pro-forma describing their policies and practices in relation to preventing and dealing with bullying incidents. Qualitative methods will examine implementation processes, operation of intervention mechanisms, the role of contextual factors and will interrogate patterns in the quantitative data. Focus groups will be conducted in intervention study schools, with teachers, school nurses who have implemented the intervention, school children who have received Support group and peers who have participated in Support group, to examine acceptability, knowledge

**Table 2. Overview of planned studies, outcome measures and time of measurement.**

| No | Population | Outcome | Outcome measure | Time of measurement | RQnoᵃ |
|---|---|---|---|---|---|
| Study 1 | All students | Bullying<br>Quality of life<br>Class environment | Elevundersøkelsen-UDIR | At baseline, 4 months in, end of intervention, and 6 and 9 months later. | pRQ1–3, sRQ8 |
| Study 2 and 3 | Participants with a peer support group | Bullying<br>Quality of life<br>Mental health<br>General self-efficacy | Olweus [II]<br>KID screen-10[I]<br>SDQ [III]<br>GSE scale [IV] | Baseline, end of intervention, and at 6 and 9 months after | pRQ4, sRQ5–8 |
| Study 4 | children, teachers, school nurses and residents | Participant perspectives on study implementation | Quantitative data<br>Qualitative data | At implementation, during intervention, end of intervention, and 6 months after. | sRQ9–12 |

ᵃCorresponding research questions,

[I]KIDSCREEN-10, [33]

[II]Olweus` questionnaire [26],

[III]Strength and Difficulties Questionnaires [34],

[IV]General self-efficacy scale [GSE] [36]

of Support group content, and whether it has shaped peer interactions. In intervention schools, we will also conduct interviews with approximately five to eight parents/ carers to explore the extent to which information about Support group has been communicated to families, and its receipt by them. Interviews and focus groups will enable us to refine understanding of key intervention mechanisms, and how contextual factors shape their operation. Qualitative findings on mechanisms – and their variation across schools, will help to refine the intervention model. Data from intervention schools will seek to identify any unintended [positive or iatrogenic] mechanisms.

## Data analyses

### Statistical analyses

The data analyses will follow Consolidated Standards of Reporting Trials (CONSORT) guidelines for RCTs. Baseline data will be presented descriptively. We will enrol about 3000 children (13 schools with number of children ranging from 1000 to 1500 in each arm) to compare the intervention and control arms.

 **a)  Children who experience being bullied and their support groups.**  We anticipate enrolling 80 children who experience being bullied. Data on children who are bullied, and their support group will be analysed longitudinally. The study unit will be each enrolled individual. The participants will be assessed at baseline (study start), after the end of the intervention and 3 and 6 months thereafter. Generalized mixed models (GLM) for repeated will be used to compare before and after the intervention outcomes on individual level for children who were targeted for support groups. The analysed outcomes will be Kids screen 10 total score, Olweus, SDQ, GSE, all considered as continuous variables. In addition, the outcome concerning being or not bullied at school [single item] will be dichotomized and analysed using GLM for binary outcomes for repeated measures.

 **b]  All enrolled children divided into an intervention and control school arms.**  GLM will also be used to analyse possible differences between the intervention and control arms for all outcomes. The study unit will be a class as due to data protection considerations we were not allowed to collect data on individual level. The individual school will be treated as clusters. Type of intervention [group], time [assessment point] and an interaction term group* time will be entered as fixed factors. The interaction term will be included to adjust for possible baseline differences between the groups. If necessary, the models will be adjusted for possible confounders [socio-demographic variables collected at study start]. School will be treated as clusters and entered as a random factor. All questionnaires will be administered at inclusion [study start], and 4 months later, after the end of the intervention and 6 months thereafter. The analysed variables will be Kid screen 10 (the total domain and sub-domains) and the Pupil questionnaire (Elevundersøkelsen). All outcomes will be

considered continuous. For continuous outcomes we will use GLM with identity link, for categorical outcomes the logit link. To adjust for statistical dependencies (the same participants will be measured several times), we will use unstructured covariance matrix which does not impose any structure on the data and all variances and covariances to be distinctly estimated. If convergence is not reached, we will use exchangeable matrix where equal variances for random effects are estimated, and one common pairwise covariance or independent covariance matrix which utilizes one unique variance parameter per random effect, and all covariances are set to zero. The results will be reported as the estimated between group changes with 95% confidence intervals.

To assess possible biases due to dropouts, we will compare responders and non-responders on selected baseline variables at all assessment time points using descriptive statistics and bivariate statistical methods (chi-square test for pairs of categorical variables and t-test or Mann Whitney Wilcoxon test for continuous data).

If any outliers are observed, we will conduct sensitivity analyses with data where such observations/classes are excluded to assess possible biases.

For the process evaluation, descriptive statistical analysis of quantitative data and content analysis of qualitative data will be carried out. To address missing data, appropriate statistical methods such as multiple imputation will be applied. This method allows for the estimation of missing values based on observed data, helping to minimize bias and preserve statistical power in the analysis. We will use the package mi as implemented in Stata ver 18. No interim analyses are planned for this trial. However, if any unforeseen safety concerns arise, or if early results strongly indicate either benefit or harm, the research team will review the data, and decisions regarding termination or modification of the trial will be made in consultation with an independent ethics committee.

All tests will be two-sided and p-values <0.05 will be considered statistically significant. Analyses will be conducted using SPSS ver 28 and Stata ver 18.

## Qualitative methods

We will utilize a qualitative approach to examine the experiences of peers and school personnel. This method allows us to explore and interpret phenomena as they are understood and expressed by the participants. Through individual interviews and focus group discussions, we aim to gather in-depth insights into key phenomena and user experiences [33]. Qualitative analyses of meaning content will be carried out by informed models for qualitative analyses as described by Kvale and Brinkmann [38] and Braun and Clarke [39]. All interviews will be semi-structured. Guides for individual interviews with peers participating in Support groups and school nurses contain 4–5 predetermined themes with follow-up questions and permit an open dialog. Peers are interviewed to share their perspectives on the Support group and their experiences within the school environment. Individual interviews with school nurses and teachers explore their insights into the intervention. The comprehensive interview guide addresses key themes such as experiences with Support groups in the school, the implementation process, and factors influencing successful implementation. Additionally, participants are asked to reflect on their experiences with establishing Support groups and the determinants critical to effective implementation.

## Ethics

Ethical guidelines for research will be followed. The Norwegian Centre for Research Data has granted ethical permission (case no 717417). Active consent will be obtained from all parents of children receiving support group intervention (children who are being bullied). Consent will be collected when the child's parents are contacted during step 2 of the intervention. Consent will also be obtained from the parents of peers who, after participating in the support group, are invited to take part in interviews. Additionally, consent will be sought from school nurses, teachers, and school leadership at both intervention and control schools participating in the interviews. These interviews are part of the process evaluation. At the

class level, informed consent will be obtained from all parents of children in grades 5–7 at both intervention and control schools. Parents will be informed about the project both verbally and in writing and will have the option to withdraw their child from participation if they wish. All participants will be informed that their participation in the study is voluntary, and they are free to choose whether they want to participate in the project. Even though a school child qualifies to participate in the study they will be assured that they can withdraw from the study at any time, and still receive care and help from the health center as usual.

Participants will continue to have access to standard school health services during and after the trial. Should any participant experience harm or distress as a result of trial participation, they will be offered appropriate support services, including psychological assistance from the school's health team. No additional compensation is planned for participants, as the trial poses minimal risk. However, any harm reported will be handled in accordance with ethical guidelines, and necessary interventions will be provided. A data management plan according to EUs privacy regulation [GDPR] will be carried out. The final trial dataset will be accessible to the principal investigators and designated data analysts. Access to the dataset will be managed in accordance with Oslo Metropolitan University's data protection and sharing policies, ensuring compliance with GDPR. There are no contractual agreements that limit access to the dataset for investigators. Any request for data access from external researchers will be evaluated on a case-by-case basis following ethical and legal guidelines. The researchers have registered the protocol for the study at ClinicalTrials.gov [NCT06578260]. Any important modifications to the trial protocol, such as changes in eligibility criteria, outcomes, or analysis methods, will be promptly communicated to all relevant parties. This includes trial investigators, ethics committees (REC/IRBs), trial registries, journals, and, if necessary, trial participants. The amendments will be recorded and referenced in the trial documentation, ensuring transparency and compliance with regulatory requirements.

### Communication of trial results

Upon completion of the trial, results will be communicated to all relevant stakeholders, including trial participants, healthcare professionals, and the broader public. The results will be published in peer-reviewed journals, and summaries of the findings will be made available to participants through written reports or meetings. In addition, the results will be registered in relevant trial databases, such as ClinicalTrials.gov. There are no publication restrictions, and findings will be shared freely and transparently.

### Public access to data and protocol

Upon completion of the trial, the full protocol and statistical code will be made available to the public through appropriate repositories, such as ClinicalTrials.gov or institutional repositories. These data will be accessible to external researchers upon reasonable request, ensuring compliance with ethical standards and participant privacy. There are no restrictions on the sharing of these data.

### Discussion

Bullying remains a significant public health issue with far-reaching implications for the mental, emotional, and social well-being of children. The "Support Group" intervention, which employs a solution-focused approach (SFA), presents a promising strategy to address this pervasive problem within primary school settings. This study, through a CRCT, aims to rigorously assess the effectiveness of this intervention in reducing bullying, enhancing the class environment and quality of life, and improving mental health and self-efficacy among schoolchildren.

General self-efficacy is often associated with how individuals cope with being bullied. For bullying victims, the level of self-efficacy, defined as the belief in one's ability to manage situations, can significantly influence their experience and

response to bullying, and children with high self-efficacy are often better equipped to handle bullying. They believe in their ability to take control of the situation, such as by seeking support from friends, teachers, or other adults, or by standing up for themselves. This can help mitigate the psychological burden that bullying typically imposes [40,41]. These findings highlight the importance of interventions aimed at strengthening self-efficacy as part of comprehensive anti-bullying programs [41]. Such interventions could empower children to not only cope more effectively with bullying but also to improve their overall mental health and well-being. Researchers have also consistently found that individuals with higher levels of subjective quality of life are better able to cope with and sustain social connections during stressful life events [e.g., bullying and social exclusion]. This is because children with higher quality of life tend to have more positive self-perceptions, stronger peer relationships, and better emotional resilience, which help them navigate challenges like bullying more effectively[42]. Improved mental health can be enhanced through interventions that raise awareness of bullying, empower children to actively participate in solutions, and improve their understanding and acceptance of emotions. The Support group intervention may equip children with strategies to maintain well-being even when facing difficult emotions [43].

The findings of this study are expected to contribute significantly to the literature on anti-bullying interventions by providing robust evidence on the efficacy of the SFA-based Support Group model. The primary outcomes, including reductions in bullying incidents, will offer insights into how structured peer support can create a safer and more supportive school atmosphere. Furthermore, the process evaluation will provide critical information on the fidelity of the intervention's implementation and the factors that facilitate or hinder its success.

This study faces several potential limitations. One significant challenge is ensuring the consistent quality of intervention implementation across all participating schools. Variability in how school personnel and school nurses engage with and execute the intervention could impact the outcomes. To mitigate this, the study includes comprehensive training for school staff and school nurses and ongoing support throughout the intervention period. However, variations in staff engagement, school culture, and student demographics may still influence the results. Despite these challenges, the study's innovative approach and comprehensive design offer significant potential to advance the field of bullying prevention. This study will provide a understanding of how and why the Support Group intervention works, and for whom it is most effective. This knowledge will be invaluable for educators, policymakers, and researchers seeking to develop more effective, evidence-based strategies to address bullying in schools.

In conclusion, this study has the potential to make a meaningful contribution to the ongoing efforts to reduce bullying and improve the mental health and well-being of children. The findings will not only inform best practices in Norway but could also serve as a model for other countries looking to implement similar interventions. The insights gained from this research will be crucial for shaping future strategies aimed at creating safer and more inclusive school environments globally.

## Author contributions

**Conceptualization:** Lisbeth Valla, Bente Sparboe-Nilsen, Lisbeth G Kvarme, Milada Hagen, Brita Askeland Winje.

**Methodology:** Lisbeth Valla, Bente Sparboe-Nilsen, Milada Hagen, Brita Askeland Winje.

**Supervision:** Milada Hagen, Brita Askeland Winje.

**Writing – original draft:** Lisbeth Valla, Lisbeth G Kvarme, Brita Askeland Winje.

**Writing – review & editing:** Lisbeth Valla, Bente Sparboe-Nilsen, Lisbeth G Kvarme, Therese Haugerud Bjerketveit, Milada Hagen, Brita Askeland Winje.

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
