## [Decision Letter · Decision Letter 0]

13 Dec 2024

PONE-D-24-51154A cluster randomised controlled trial investigating the effectiveness of the‘Support group’ intervention in primary schools in Norway: a study protocolPLOS ONE

Dear Dr. Valla,

Thank you for submitting your manuscript to PLOS ONE. After careful consideration, we feel that it has merit but does not fully meet PLOS ONE’s publication criteria as it currently stands. Therefore, we invite you to submit a revised version of the manuscript that addresses the points raised during the review process.

We look forward to receiving your revised manuscript.

Kind regards,

Alejandro Botero Carvajal, MD

Academic Editor

PLOS ONE

2. Please remove your figures from within your manuscript file, leaving only the individual TIFF/EPS image files, uploaded separately. These will be automatically included in the reviewers’ PDF.

3. Please ensure that you refer to Figure 3 in your text as, if accepted, production will need this reference to link the reader to the figure.

Reviewers' comments:

Reviewer's Responses to Questions

**Comments to the Author**

1. Does the manuscript provide a valid rationale for the proposed study, with clearly identified and justified research questions?

Reviewer #1: Partly

Reviewer #2: Yes

2. Is the protocol technically sound and planned in a manner that will lead to a meaningful outcome and allow testing the stated hypotheses?

Reviewer #1: Partly

Reviewer #2: Yes

3. Is the methodology feasible and described in sufficient detail to allow the work to be replicable?

Reviewer #1: Yes

Reviewer #2: Yes

4. Have the authors described where all data underlying the findings will be made available when the study is complete?

Reviewer #1: Yes

Reviewer #2: No

5. Is the manuscript presented in an intelligible fashion and written in standard English?

Reviewer #1: Yes

Reviewer #2: Yes

6. Review Comments to the Author

You may also provide optional suggestions and comments to authors that they might find helpful in planning their study.

Reviewer #1: This manuscript presents a study protocol for a cluster randomized controlled trial (CRCT) investigating the effectiveness of a support group intervention for bullying in Norwegian primary schools. While the study addresses an important topic and the chosen methodology is generally appropriate, several aspects of the protocol require clarification and strengthening before the study commences. The rationale for the intervention is well-articulated, but the description of the intervention itself lacks sufficient detail.

The description of the Support Group intervention is too general. While the principles of the solution-focused approach (SFA) are mentioned, the specific activities and procedures within the support group sessions are not clearly outlined.

Table 1 provides a basic overview, but it lacks detail regarding the content and duration of each step. How are the "suggestions to help the bullied child" generated and implemented? How is progress monitored and evaluated within the SFA framework? A more detailed manual or protocol for the intervention is needed to ensure fidelity and replicability.

While the manuscript lists the chosen outcome measures, it lacks a detailed explanation of how these measures will be used to answer the research questions. For instance, how will the Olweus questionnaire be used to measure a reduction in bullying (pRQ1)? Will changes in scores be analyzed, or specific items related to bullying frequency? Similarly, which specific domains of the KIDscreen-27 and SDQ will be used to address the research questions related to quality of life and mental health? Clearer links between the research questions and the specific aspects of the outcome measures are needed.

The analysis plan is too brief. While generalized mixed models (GLM) are mentioned, the specific models to be used for each research question are not specified. Which covariates will be included in the models? How will the clustering effect be accounted for in the analysis? Will interaction effects between intervention type and time be explored? A more detailed description of the statistical analysis plan, including the specific models and variables, is required. The rationale for using an unstructured covariance matrix should also be justified. Given the longitudinal nature of the data, exploring alternative covariance structures might be warranted. Furthermore, the plan to address missing data using multiple imputation is appropriate, but the specific imputation model and assumptions should be described.

The process evaluation plan is a strength of the protocol, but it could be further developed. While the manuscript mentions collecting data on fidelity, acceptability, and contextual factors, the specific methods for collecting these data are not clearly described. What specific questions will be asked in the focus groups and interviews? How will the qualitative data be analyzed? A more detailed description of the data collection and analysis methods for the process evaluation is needed. Furthermore, how will the findings from the process evaluation be integrated with the outcome evaluation to understand the intervention's effectiveness?

The rationale for the sample size calculation at the individual child level needs further clarification. The manuscript mentions expecting a change of 3 points in the "social support and peers" domain of KIDscreen-27. However, the basis for this assumption is not provided. Providing a reference or justification for this expected change would strengthen the sample size calculation.

While the rationale for using "usual care" as the control is provided, a more detailed description of what "usual care" entails in the participating schools is needed.

The manuscript mentions informing parents about the support groups. However, it is unclear whether active consent will be obtained from parents for their children to participate in the support groups. Clarifying the consent process for both the overall study and the support groups is important.

While blinding of participants and intervention providers is not feasible, the manuscript should clarify whether the individuals conducting the baseline and follow-up assessments will be blinded to group allocation.

Overall comments: Providing more detail about the intervention, outcome measures, analysis plan, and process evaluation will significantly strengthen the protocol and enhance the rigor and interpretability of the study's findings. This study has the potential to contribute valuable knowledge to the field of bullying prevention, and addressing these concerns will ensure that the study is conducted to the highest scientific standards.

Reviewer #2: 1. I would encourage authors to provide a conceptual framework for this support group intervention addressing bullying prevention.

Suggestions that you can consider what to include: Theory (Bronfenbrenner’s Ecological Systems Theory) → Components (which construct from theory will work on which component) → Mediators/causal factors → Outcomes (both proximal and distal outcome).

2. Add a separate section for ‘Measures’ for each of the scales to be used for this trial. Are all those scales validated in Norwegian?

7. PLOS authors have the option to publish the peer review history of their article (what does this mean? ). If published, this will include your full peer review and any attached files.

**Do you want your identity to be public for this peer review?** For information about this choice, including consent withdrawal, please see our Privacy Policy .

Reviewer #1: No

Reviewer #2: No

---

## [Author Response · Author response to Decision Letter 0]

11 Feb 2025

Reviewer #1:

Comment 1:

This manuscript presents a study protocol for a cluster randomized controlled trial (CRCT) investigating the effectiveness of a support group intervention for bullying in Norwegian primary schools. While the study addresses an important topic and the chosen methodology is generally appropriate, several aspects of the protocol require clarification and strengthening before the study commences. The rationale for the intervention is well-articulated, but the description of the intervention itself lacks sufficient detail.

The description of the Support Group intervention is too general. While the principles of the solution-focused approach (SFA) are mentioned, the specific activities and procedures within the support group sessions are not clearly outlined.

Response 1: Thank you for your valuable comment. We found your feedback very helpful and completely agree with your observation. In response, we have provided a more detailed description of the intervention on page 12, lines 286–301, and page 13 lines 302-324. We hope this addition enhances the manuscript and addresses your concern effectively.

Comment 2:

Table 1 provides a basic overview, but it lacks detail regarding the content and duration of each step. How are the "suggestions to help the bullied child" generated and implemented? How is progress monitored and evaluated within the SFA framework? A more detailed manual or protocol for the intervention is needed to ensure fidelity and replicability.

Response 2: Thank you for your feedback. As mentioned in our response to your first comment, we agree that providing more detail is essential. We have now included a more comprehensive description of the intervention, detailing the content and duration of each step, the generation and implementation of "suggestions to help the bullied child," and the monitoring and evaluation process within the SFA framework. This additional information can be found on page 12 and 13, lines 286-324 and in table 1. We hope this revision clarifies the protocol and enhances the manuscript's fidelity and replicability.

Comment 3:

While the manuscript lists the chosen outcome measures, it lacks a detailed explanation of how these measures will be used to answer the research questions. For instance, how will the Olweus questionnaire be used to measure a reduction in bullying (pRQ1)? Will changes in scores be analyzed, or specific items related to bullying frequency? Similarly, which specific domains of the KIDscreen-27 and SDQ will be used to address the research questions related to quality of life and mental health? Clearer links between the research questions and the specific aspects of the outcome measures are needed.

Response 3: We appreciate your suggestion regarding the need for a clearer explanation of how the outcome measures will be used to address the research questions. A more detailed description of each instrument has been added on page 15 and 16, line 359-382.

We sincerely apologize for the incorrect information in the previous draft of our manuscript. The correct instrument used is KIDSCREEN-10, and this has now been corrected.

Comment 4:

The analysis plan is too brief. While generalized mixed models (GLM) are mentioned, the specific models to be used for each research question are not specified. Which covariates will be included in the models? How will the clustering effect be accounted for in the analysis? Will interaction effects between intervention type and time be explored? A more detailed description of the statistical analysis plan, including the specific models

and variables, is required. The rationale for using an unstructured covariance matrix should also be justified. Given the longitudinal nature of the data, exploring alternative covariance structures might be warranted. Furthermore, the plan to address missing data using multiple imputation is appropriate, but the specific imputation model and assumptions should be described.

Response 4:

We have elaborated greatly on the description of statistical methods and all the methods and statistical models are now clearly specified. Please see the revised version of the manuscript/ Data analysis section, page 19-20, lines 422-472 All models will include fast effect (type of intervention and assessment time) and an interaction term – as suggested by the reviewer. We will also include a random term to adjust for clustering. We do not want to impose any structure on our data; therefore we intend to use unstructured covariance matrix. However, if convergence is not achieved, we will explore alternative covariance structures.

Regarding multiple imputation, we will use the package ‘mi’ for multiple imputation as implemented in Stata ver 18. It is a model based multiple imputation methods. A reference to this approach is added to the manuscript (STATA MULTIPLE IMPUTATION REFERENCE MANUAL, RELEASE 18. Stata Press, 4905 Lakeway Drive, College Station, Texas 77845 ISBN-10: 1-59718-391-1 and ISBN-13: 978-1-59718-391-8)

Comment 5:

The process evaluation plan is a strength of the protocol, but it could be further developed. While the manuscript mentions collecting data on fidelity, acceptability, and contextual factors, the specific methods for collecting these data are not clearly described. What specific questions will be asked in the focus groups and interviews? How will the qualitative data be analyzed? A more detailed description of the data collection and analysis methods for the process evaluation is needed. Furthermore, how will the findings from the process evaluation be integrated with the outcome evaluation to understand the intervention's effectiveness?

Response 5:

Thank you for highlighting the importance of further elaboration on the process evaluation plan. We acknowledge your concerns and have addressed them as follows:

Data Collection Methods: For fidelity, acceptability, and contextual factors, we will employ both quantitative and qualitative approaches. Quantitative methods include termly checklists completed by schools to document adherence to the Support Group strategies and practices. Qualitative methods involve individual and focus groups interviews with teachers, school nurses, peers participated in a Support group. All interviews will be semi-structured. Guides for individual interviews with peers participating in Support groups and school nurses contain 4–5 predetermined themes with follow-up questions and permit an open dialog. Peers are interviewed to share their perspectives on the Support group and their experiences within the school environment. Individual interviews with school nurses and teachers explore their insights into the intervention. The comprehensive interview guide addresses key themes such as experiences with Support groups in the school, the implementation process, and factors influencing successful implementation. Additionally, participants are asked to reflect on their experiences with establishing Support groups and the determinants critical to effective implementation.

We will utilize a qualitative approach to examine the experiences of peers and school personnel. This method allows us to explore and interpret phenomena as they are understood and expressed by the participants. Through individual interviews and focus group discussions, we aim to gather in-depth insights into key phenomena and user experiences. Qualitative analyses of meaning content will be carried out by informed models for qualitative analyses as described by Kvale and Brinkmann (2009) and Braun and Clarke (2022).

This is now described in detail in the manuscript on page 20 line 470-484.

Comment 6:

The rationale for the sample size calculation at the individual child level needs further clarification. The manuscript mentions expecting a change of 3 points in the "social support and peers" domain of KIDscreen-27. However, the basis for this assumption is not provided. Providing a reference or justification for this expected change would strengthen the sample size calculation. While the rationale for using "usual care" as the control is provided, a more detailed description of what "usual care" entails in the participating schools is needed.

Response 6:

We acknowledge the need for further clarification regarding the anticipated change of 3 points in the domain of KIDSCREEN-10 as clinically relevant. This assumption was based on findings from previous studies aimed at improving social well-being in children (Ravens-Sieberer et al 2019).

In our study, 'usual care' refers to the standard support and interventions that schools typically provide to address bullying and promote students' psychosocial well-being. This may include structured anti-bullying plans, regular student well-being surveys, teacher-student relationship building, parental collaboration, and student counselling. Specialized teams may assist in severe cases. We have clarified this in the revised manuscript to ensure transparency regarding the control condition on page 11 line 277-280).

Comment 7:

The manuscript mentions informing parents about the support groups. However, it is unclear whether active consent will be obtained from parents for their children to participate in the support groups. Clarifying the consent process for both the overall study and the support groups is important.

Response 7:

Thank you for your valuable feedback. We recognize the importance of providing clarity on the consent process for both the overall study and participation in the support groups. To address this, we have included the relevant details on page 20, lines 490–499.

Active consent will be obtained from all parents of children receiving support group intervention (children who are being bullied). Consent will be collected when the child's parents are contacted during step 2 of the intervention. Consent will also be obtained from the parents of peers who, after participating in the support group, are invited to take part in interviews. Additionally, consent will be sought from school nurses, teachers, and school leadership at both intervention and control schools participating in the interviews. These interviews are a part of the process evaluation. At the class level, informed consent will be obtained from all parents of children in grades 5–7 at both intervention and control schools. Parents will be informed about the project both verbally and in writing and will have the option to withdraw their child from participation if they wish.

Comment 8:

While blinding of participants and intervention providers is not feasible, the manuscript should clarify whether the individuals conducting the baseline and follow-up assessments will be blinded to group allocation.

Response 8:

The manuscript specifies on page 9, line 220, that blinding is inherently limited due to the nature of the intervention. Participants and school staff responsible for delivering the intervention cannot be blinded because of the visible and interactive aspects of the Support Group intervention. Furthermore, the baseline and follow-up assessments are also conducted by school staff, although not by the same individuals delivering the intervention. Nevertheless, these staff members are not blinded to group allocation. To mitigate potential biases in the evaluation of results, outcome assessors and data analysts will remain blinded to group assignments. If unblinding becomes necessary at any stage, it will follow strict procedural guidelines to ensure that the integrity and validity of the trial are preserved.

We have added a sentence on page 9 stating that participants and school staff involved in delivering the intervention, as well as conducting the baseline and follow-up assessments, cannot be blinded due to the visible nature of the intervention.

Comment 9: Overall comments:

Providing more detail about the intervention, outcome measures, analysis plan, and process evaluation will significantly strengthen the protocol and enhance the rigor and interpretability of the study's findings. This study has the potential to contribute valuable knowledge to the field of bullying prevention, and addressing these concerns will ensure that the study is conducted to the highest scientific standards.

Response 9:

Thank you for your feedback. We have made several revisions to enhance the rigor of the protocol and to ensure that the study’s findings are more robust and interpretable. Specifically, we have expanded the description of the intervention, outcome measures, analysis plan (the Statistical analysis section has been greatly expanded), and process evaluation. We have also elaborated on how the primary and secondary outcomes, such as the Olweus Questionnaire and KIDSCREEN-10, align with the study objectives and their psychometric properties, including validity and reliability. In the analysis plan, we have added details on how clustering effects will be managed and missing data handled. These updates aim to strengthen the scientific rigor of the study and ensure it contributes meaningful knowledge to the field of bullying prevention.

Reviewer #2:

Comment 1:

I would encourage authors to provide a conceptual framework for this support group intervention addressing bullying prevention.

Suggestions that you can consider what to include: Theory (Bronfenbrenner’s Ecological Systems Theory) → Components (which construct from theory will work on which component) → Mediators/causal factors → Outcomes (both proximal and distal outcome).

Response 1:

Thank you for your feedback. We completely agree with the reviewer that Bronfenbrenner’s Ecological Systems Theory is well-suited to the support group intervention. Bronfenbrenner’s theory is mentioned on page 5, but based on your feedback, we have now added which constructs from the theory are utilized in which specific components, as well as mediators/causal mechanisms and outcomes (both proximal and distal outcomes). By incorporating this comprehensive explanation on page 5, we believe that we have not addressed the reviewer’s suggestion and enhanced the theoretical and academic rigor- of our manuscript.

Bullying is conceptualized as a complex social phenomenon influenced by interactions within the microsystem (e.g., peers and school), mesosystem (e.g., school and family relationships), exosystem (e.g., school policies), and macrosystem (e.g., societal and cultural norms).

The intervention draws on this theory by targeting specific ecological levels to foster change. At the microsystem level, the Support Group intervention enhances peer relationships and creates a supportive social network to address the child’s immediate social context. The mesosystem is engaged by strengthening collaboration between school staff and families, promoting a shared commitment to bullying prevention. At the exosystem level, the intervention aligns with school policies to integrate systemic approaches to bullying prevention. The macrosystem is indirectly influenced by raising awareness of bullying as a broader societal issue, fostering inclusivity and social responsibility. Mediators such as improved peer support, enhanced empowerment through self-efficacy, and the creation of a positive classroom environment are critical pathways through which the intervention achieves its effects. These mechanisms are hypothesized to lead to proximal outcomes, such as reduced bullying incidents and improved peer relationships. Over time, these proximal changes are expected to translate into distal outcomes, including enhanced mental health, quality of life, and general self-efficacy of the participating children.

This conceptual framework illustrates how the theoretical basis informs the design and implementation of the Support Group intervention, providing a clear link between the intervention components, mediators, and desired outcomes.

Comment 2:

Add a separate section for ‘Measures’ for each of the scales to be used for this trial. Are all those scales validated in Norwegian?

Response 2:

Thank you for your feedback. We have added a section Instruments and methods on page 16 and 17 where a detailed overview of each scale used in this trial is included. Additionally, we have

---

## [Decision Letter · Decision Letter 1]

15 Apr 2025

A cluster randomised controlled trial investigating the effectiveness of the

‘Support group’ intervention in primary schools in Norway: a study protocol

PONE-D-24-51154R1

Dear Dr. Valla,

We’re pleased to inform you that your manuscript has been judged scientifically suitable for publication and will be formally accepted for publication once it meets all outstanding technical requirements.

Kind regards,

Alejandro Botero Carvajal, MD

Academic Editor

PLOS ONE

Additional Editor Comments (optional):

Reviewers' comments:

Reviewer's Responses to Questions

**Comments to the Author**

1. Does the manuscript provide a valid rationale for the proposed study, with clearly identified and justified research questions?

Reviewer #3: Yes

Reviewer #4: Yes

2. Is the protocol technically sound and planned in a manner that will lead to a meaningful outcome and allow testing the stated hypotheses?

Reviewer #3: Yes

Reviewer #4: Yes

3. Is the methodology feasible and described in sufficient detail to allow the work to be replicable?

Reviewer #3: Yes

Reviewer #4: Yes

4. Have the authors described where all data underlying the findings will be made available when the study is complete?

Reviewer #3: Yes

Reviewer #4: Yes

5. Is the manuscript presented in an intelligible fashion and written in standard English?

Reviewer #3: Yes

Reviewer #4: Yes

6. Review Comments to the Author

You may also provide optional suggestions and comments to authors that they might find helpful in planning their study.

Reviewer #3: I encourage the author to carry out more in-depth research, which is very meaningful for reducing campus bullying.

Reviewer #4: I consider that the study meets the fundamental requirements to be considered a good work and the comments and their respective proposals for improvement raised by the people who have carried out the review have been taken into account.

7. PLOS authors have the option to publish the peer review history of their article (what does this mean? ). If published, this will include your full peer review and any attached files.

**Do you want your identity to be public for this peer review?** For information about this choice, including consent withdrawal, please see our Privacy Policy .

Reviewer #3: No

Reviewer #4: No

---

## [Editor Report · Acceptance letter]

PONE-D-24-51154R1

PLOS ONE

Dear Dr. Valla,

I'm pleased to inform you that your manuscript has been deemed suitable for publication in PLOS ONE. Congratulations! Your manuscript is now being handed over to our production team.

Kind regards,

on behalf of

Dr. Alejandro Botero Carvajal

Academic Editor

PLOS ONE